# A Lightweight Object Detection Method in Aerial Images Based on Dense Feature Fusion Path Aggregation Network

Liming Zhou [1,2] , Xiaohan Rao [1,2] , Yahui Li [1,2], Xianyu Zuo [1,2,*], Baojun Qiao [1,2] and Yinghao Lin [1,2]

[1] Henan Key Laboratory of Big Data Analysis and Processing, Henan University, Kaifeng 475000, China; lmzhou@henu.edu.cn (L.Z.); rshenhaibb@henu.edu.cn (X.R.); liyahui@henu.edu.cn (Y.L.); qbj@henu.edu.cn (B.Q.); linyh@henu.edu.cn (Y.L.)
[2] School of Computer and Information Engineering, Henan University, Kaifeng 475000, China
[*] Correspondence: xianyu_zuo@henu.edu.cn

**Abstract:** In recent years, significant progress has been obtained in object detection using Convolutional Neural Networks (CNNs). However, owing to the particularity of Remote Sensing Images (RSIs), common object detection methods are not well suited for RSIs. Aiming at the difficulties in RSIs, this paper proposes an object detection method based on the Dense Feature Fusion Path Aggregation Network (DFF-PANet). Firstly, for better improving the detection performance of small and medium-sized instances, we propose Feature Reuse Module (FRM), which can integrate semantic and location information contained in feature maps; this module can reuse feature maps in the backbone to enhance the detection capability of small and medium-sized instances. After that, we design the DFF-PANet, which can help feature information extracted from the backbone to be fused more efficiently, and thus cope with the problem of external interference factors. We performed experiments on the Dataset of Object deTection in Aerial images (DOTA) dataset and the HRSC2016 dataset; the accuracy reached 71.5% mAP, which exceeds most object detectors of one-stage and two-stages at present. Meanwhile, the size of our model is only 9.2 M, which satisfies the requirement of being lightweight. The experimental results demonstrate that our method not only has better detection accuracy but also maintains high efficiency in RSIs.

**Keywords:** feature reuse module; residual dense block; dense feature fusion; remote sensing

## 1. Introduction

With the progress of RSIs sensors, people can obtain high-quality and high-resolution aerial images by using remote sensing technology. Meanwhile, target detection in RSIs is also of great significance in military, civil and other aspects. Nowadays, deep learning has promoted great progress in various computer vision problems, for instance, object classification [1–3], object detection [4–6], object tracking [7,8]. The application of deep learning models to aerial object detection has aroused more and more attention.

For the past few years, CNNs has emerged in many object detection algorithms, which have obtained good results both in speed and accuracy. Compared with traditional object detection methods, for example, Deformable Parts Model (DPM), Histogram of Oriented Gradients (HOG) and Support Vector Machine (SVM), object detection frameworks based on CNNs make up for two issues of traditional ground detection [9]. One is limited coverage the other is lack of detection data. With its strong feature extraction ability and feature representation ability, it has made great achievements in object detection. Among them, You Only Look Once (YOLO) is a typical single-stage algorithm. In 2016, the YOLO detection model [10] was proposed by Redmon et al., which directly classifies the input images to predict. The speed of detection is faster than the previous models but at the cost of poor detection performance. In 2017, Redmon et al. proposed the YOLOv2 detection model with Darknet-19 as the backbone [11], which improves detection performance. In 2018,

Redmon et al. proposed the YOLOv3 detection model [12], which utilizes multi-scales to extract rich features of different resolutions, greatly improving the accuracy of small targets. In 2019, Bochkovskiy the YOLOv4 detection model with Cross Stage Partial Darknet 53 (CSPDarknet53) as the backbone [13] to enhance the speed of detection and ensure the network accuracy. Subsequently, the YOLOv5 detection model has aroused the attention of a wide range of scholars on account of its advantages of high speed and high precision.

Although the above-mentioned detection models have attained good results, owing to the particularity of aerial images, ordinary object detectors are not well suited for RSIs. Compared with target detection in natural imagery, for instance, the Pascal Visual Object Classes (Pascal VOC) dataset [14] and Microsoft Common Objects in Context (MS COCO) dataset [15], object detection in RSIs usually faces the following challenges.

1.  The aerial images are generally of a large size, leading to the result that the size of targets is small relative to the imagery, which is easy to produce missed detection.
2.  RSIs are often interfered with by external causes, such as shadows, similar instances and complex backgrounds, making it hard to distinguish texture rules between objects and false objects.
3.  When some instances are placed side by side in RSIs, Non-Maximum Suppression (NMS) will filter bounding boxes of different objects, resulting in missed detection.

The difficulties of object detection in RSIs are demonstrated in Figure 1.

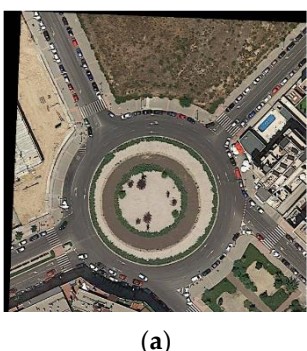 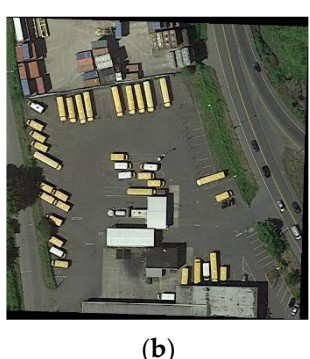 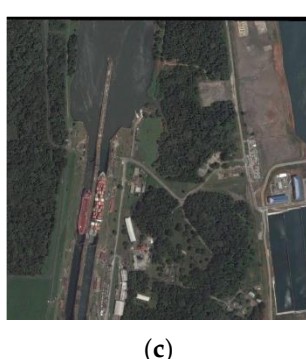

(**a**)                    (**b**)                    (**c**)

**Figure 1.** The difficulties of object detection in RSIs. (**a**) Represents the small size of remote sensing objects; (**b**) represents the interference of RSIs by similar objects and other external factors; (**c**) represents the side-by-side placement of remote sensing objects.

To solve the problems mentioned above, researchers put forward corresponding solutions. For instance, to promote the multi-scale detection capability of the network, Yuan et al. [16] put forward an end-to-end Multi-Feature Pyramid Network (MFPNet), which combines global semantic features and local detail features by constructing the multi-feature pyramid module. However, the network has certain limitations for small object detection when carrying out rotated object detection. With the purpose of solving the complex and changeable high-resolution RSIs better, Huang et al. [17] proposed Multi-scale Feature Fusion and Cross-Scale Feature Fusion of Multi-level Pyramid Network (CF2PN) based on a Multi-level and Multi-scale Detector (M2Det). However, the performance of the network is still not ideal for more complex backgrounds and fuzzy aerial images. Zhu et al. [18] designed Multi-scale SELU DenseNet (MSE-DenseNet) and promoted the anchor allocation strategy to handle the problem of the large difference in object scale. However, the network cannot reach satisfactory results when carrying out the fine and three-dimensional RSIs detection task. For optimizing the detection performance of the YOLO algorithm in RSIs, Qu et al. [6] designed the YOLOv3 model with an auxiliary network. However, the detection speed is not ideal. Aimed at solving the problem of information loss due to down-sampling and the unsatisfactory efficiency of existing object detectors in RSIs, Zhang et al. [19] proposed an end-to-end object detector of RSIs based on the improved YOLO algorithm, thus improving object detection efficiency in complex scenes. However, there is still room for improvement in extracting and combining contextual information.

To deal with the aforementioned problems, this article raises a lightweight object detection method, which has high computational efficiency. The contributions made in this thesis can be summarized as follows.

1.  This article proposes an object detection method for aerial images. This method is not only lightweight but can also carry out accurate and efficient detection work in RSIs.
2.  In order to strengthen the ability of the model to detect small and medium-sized objects, semantic and location information in feature maps is fused by the Feature Reuse Module (FRM), which can enrich feature information extracted from the backbone.
3.  A Dense Feature Fusion Path Aggregation Network (DFF-PANet) by using Cross Stage Residual Dense Block (CSRDB) has been designed to handle the problem of external interference caused by complex and changeable RSIs better.
4.  This study uses the DOTA and the HRSC2016 datasets for experiments to validate the model we put forward and then analyzes the effects of every improvement we suggested through a series of comparative and ablation experiments.

The rest of this article consists of the following parts. Section 2 reviews the object detection algorithms and feature pyramid. In Section 3, the network suggested in this thesis is depicted in detail. In Section 4, we present the results of the experiments on the DOTA and the HRSC2016 datasets. Section 5 discusses the proposed method. Section 6 briefly summarizes the results of this thesis.

## 2. Related Works

This section presents the existing object detection algorithms and the related knowledge of the feature pyramid in brief.

### 2.1. Object Detection Algorithms

CNN-based object detection algorithms can be separated into two categories [20]. One is object detection algorithms based on anchor-box, which generates a variety of bounding boxes and labels. Moreover, the number of bounding boxes should be large enough to ensure sufficient overlap between bounding boxes and ground truth boxes. Object detection algorithms based on anchor-box are divided into one-stage and two-stage object detection algorithms. One-stage object detectors are fast, but the accuracy is not as good as two-stage ones. While in two-stage object detectors, the step of Region of Interest (RoI) extraction makes the detection accuracy high, but the speed is inferior to that of single-stage ones. The other is object detection algorithms based on anchor-free, the candidate box in the candidate region method and regression method is eliminated, then high-quality anchor boxes are generated. The advantages and disadvantages of different object detection algorithms are shown in Table 1.

### 2.2. Feature Pyramid

Feature pyramid is widely used in object detection networks to detect instances of different scales. The pyramid network structure with different features [21] is shown in Figure 2. A Single Shot MultiBox Detector (SSD) [22] makes predictions through feature maps of different resolutions generated in the backbone (as shown in Figure 2a, Pyramid Feature Hierarchy). However, different levels cause the issue of the semantic gap. Shallow feature maps have high resolution but lack rich semantic information [23]. The detection performance of small targets is poor. The feature pyramid network [23] fully combines semantic information of deep feature maps and shallow ones by introducing a top-down channel and a horizontal connection (as shown in Figure 2b, Feature Pyramid Network). However, multi-layer feature layer fusion not only brings high precision but also brings a large amount of computation. The path aggregation network [24] improves the utilization rate of low-level feature information by adding bottom-up paths so as to increase the transmission efficiency of low-level information [24] (as shown in Figure 2c, Path Aggregation Network). However, there is still information loss during feature information extraction in the backbone, and the utilization rate of feature information in the backbone needs to be

improved. The multi-level feature reuse module proposed in reference [21] (as shown in Figure 2d, Multi-level Feature Reuse Module) enhances the feature information expression ability of the model by reusing deep feature maps. However, its detection efficiency for medium targets needs to be improved. Although the above network structures still have some problems to be handled, they play a critical role in the performance of multi-scale object detectors and provide some inspiration for our future work.

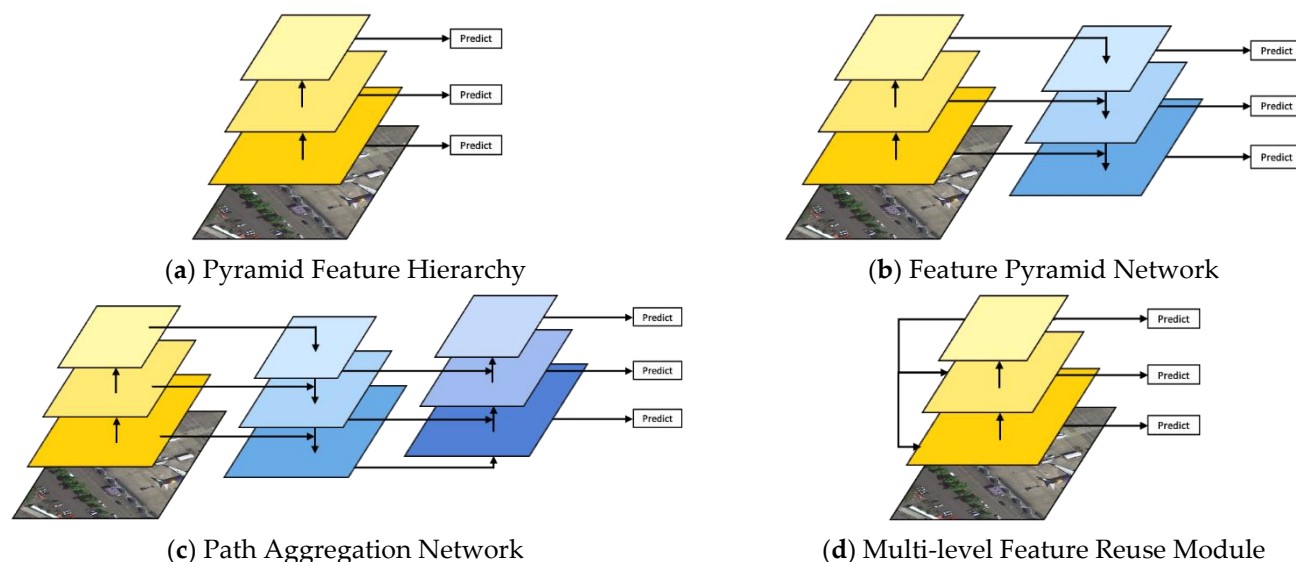

(**a**) Pyramid Feature Hierarchy

(**b**) Feature Pyramid Network

(**c**) Path Aggregation Network

(**d**) Multi-level Feature Reuse Module

**Figure 2.** Different feature pyramid network structures. (**a**) Predict using feature maps of different sizes; (**b**) combining information from top and bottom feature maps; (**c**) adding top-down and bottom-up paths; (**d**) multi-level feature reuse pyramid.

**Table 1.** The advantages and disadvantages of different object detection algorithms.

| Presence of Anchor Box | Stages | Detection Method | Advantages | Disadvantages |
|---|---|---|---|---|
| Object detection algorithms based on anchor-box | One-stage/Regression-based object detection algorithms | YOLO [10] | Object detection task is transformed into a regression problem, which highly speeds up the detection. | The position is not accurate, and the detection effect of small and dense instances is not efficient. |
| | | SSD [22] | Multi-scale detection is realized by using feature layers of different scales extracted from the backbone, and the speed of detection is fast. | Due to the deep convolutional layer, the extracted features may be lost for smaller targets. |
| | | RetinaNet [25] | Focal loss is introduced to solve the issue of positive and negative sample imbalance effectively. | Detection speed is average. |
| | Two-stage/Region-based object detection algorithms | R-CNN [26] | Extract and learn features from CNNs automatically and accelerate feature extraction. | It takes a long time to acquire regional targets. Furthermore, feature extraction is complex. |
| | | Fast R-CNN [27] | Detection efficiency is greatly improved, and training speed is significantly enhanced. | End-to-end detection is preliminarily implemented and restricted by selective search algorithms. |
| | | Faster R-CNN [28] | Region Proposal Network (RPN) is used rather than selective search algorithms to improve detection speed. | The training is divided into two stages, the region generation stage and the detection stage, which is slow and cannot satisfy the requirement of real-time. |

**Table 1.** *Cont.*

| Presence of Anchor Box | Stages | Detection Method | Advantages | Disadvantages |
|---|---|---|---|---|
| Object detection algorithms based on anchor-free | - | CornerNet [29] | By predicting the upper left and lower right corner of the object, object detection is regarded as key point detection, and the speed of detection is improved. | Easy to generate error anchor boxes. |
| | - | FCOS [30] | Many positive samples are obtained, and the problem of poor learning ability caused by a small number of positive samples are alleviated. | Semantic ambiguity may occur due to the overlapping of ground truth boxes during detection. |

## 3. Method

In this section, we will recommend our improved network structure in detail. We will describe the following four aspects. (1) Overall Network Structure; (2) YOLOv5s Backbone; (3) Dense Feature Fusion Path Aggregation Network; (4) YOLO Head.

### 3.1. Overall Network Structure

The overall network structure of our detection method in this article is depicted in Figure 3. The network is made up of three parts, YOLOv5s Backbone for feature extraction, the Dense Feature Fusion Path Aggregation Network (DFF-PANet) for feature fusion and YOLO Head for detection. Firstly, extract feature information of the input images via the backbone. To detect different size of the objects, the backbone outputs several different resolutions of feature maps ($64 \times 64$, $32 \times 32$, $16 \times 16$ and $8 \times 8$, respectively). Among them, we choose three output feature maps of the backbone as the input of the feature fusion network. Next, the extracted feature information of different sizes is sent to the DFF-PANet for feature fusion, which can enrich the feature information. Finally, these feature layers will be sent to YOLO Head for detection.

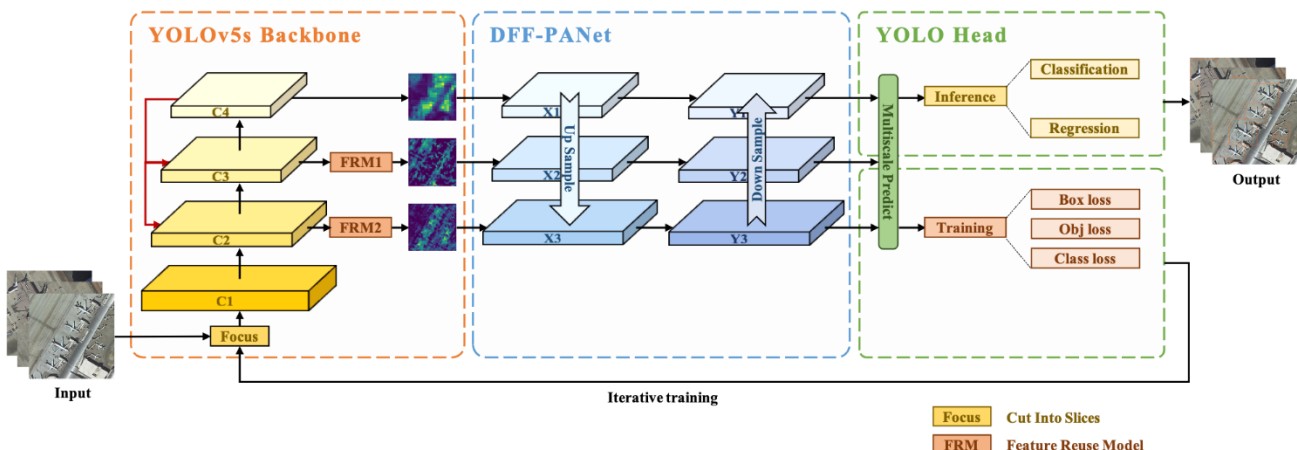

**Figure 3.** Overall network structure.

The network structure setting is illustrated in Table 2. In the backbone, $Input \in R^{3 \times 256 \times 256}$ represents the input image. In this article, the three-dimensional tensor is expressed as $X \in R^{C \times H \times W}$, where $C$, $H$ and $W$ represent the channel dimension, height and width of the feature map. The input image is sliced by Focus. After Focus, the feature map with a size of $C \in R^{32 \times 128 \times 128}$ is generated. Then, $C$ is extracted from the backbone and generates four feature maps with a resolution of $C1 \in R^{64 \times 64 \times 64}$, $C2 \in R^{128 \times 32 \times 32}$, $C3 \in R^{256 \times 16 \times 16}$ and $C4 \in R^{512 \times 8 \times 8}$, respectively. Among them, feature maps $C3$ and $C4$ are selected to fuse and generate the feature map with a size of $C3' \in R^{256 \times 16 \times 16}$ by the Feature Reuse Module (FRM),

feature maps *C2*, *C3* and *C4* are selected to fuse and generate the feature map with a size of $C2' \in R^{128 \times 32 \times 32}$ by the FRM. The FRM will be depicted in Section 3.2. In the DFF-PANet, *C4*, *C3'* and *C2'* are selected to be sent into the neck for feature fusion (the feature map, marked in red in the table). It is noteworthy that *C4* refers to the last feature map extracted from the backbone, while *C3'* and *C2'* refer to the feature maps obtained after the FRM. During feature fusion, the network will repeatedly fuse from the top-down and bottom-up to obtain *Y1*, *Y2* and *Y3* of different resolutions and send them to YOLO Head for prediction. In YOLO Head, it is divided into the inference and training stages. If it is the inference stage, classification and regression are carried out to obtain the final output. If it is the training stage, the loss will be calculated, and iterative training is conducted until the loss value no longer decreases, and the training tends to be stable.

**Table 2.** The network structure setting. The red part represents the feature maps to be sent into DFF-PANet for feature fusion.

| | Network Module | | Input | Output | Operation |
|---|---|---|---|---|---|
| Backbone | Focus | | *Input* | *C* | Slice |
| | Extract Feature | | *C* | *C1*, *C2*, *C3*, *C4* | Convolution |
| | FRM | FRM1 | FRM (*C3*, *C4*) | *C3'* | Fusion |
| | | FRM2 | FRM (*C2*, *C3*, *C4*) | *C2'* | Fusion |
| DFF-PANet | Top-down path | | *C4, C3', C2'* | *X1, X2, X3* | Fusion |
| | Bottom-up path | | *X1, X2, X3* | *Y1, Y2, Y3* | Fusion |
| YOLO Head | Inference | Classification | | - | |
| | | Regression | | - | |
| | Training | Box loss | | Complete Loss (CIoU) | |
| | | Obj loss | | BCEWithLogitsLoss | |
| | | Class loss | | BCEWithLogitsLoss | |

### 3.2. YOLOv5s Backbone

The YOLOv5 model can be divided into five models, YOLOv5n, YOLOv5s, YOLOv5m, YOLOv5l and YOLOv5x. While in this network, YOLOv5s is used in the backbone for subsequent enhancement, which not only has fewer network parameters but also can maintain high accuracy with high speed.

However, since low-level and middle-level feature maps contain less semantic information, these feature maps of information processing may influence the performance of small and medium object detection. It is of great significance to promote the detection accuracy of small and medium objects for ensuring the balance of semantic information between low-level and high-level feature maps. Inspired by reference [21], we adopted the Feature Reuse Module (FRM) in the backbone, which provides an efficient reuse mechanism for the backbone. The FRM is depicted in Figure 4. The mathematical expression of the FRM can be expressed as:

$$FRM_t = \psi_t\{x, T_i(x_i)\}, \; x_i \in S \tag{1}$$

$$y_i = \Psi_r\{y_{i-1}, \; FRM_t\}, \; r \in R \tag{2}$$

where *x* represents the *C1* feature map in the backbone after Focus. *S* represents the feature maps to be reused in the backbone (*C2* and *C3*, respectively), which is called the source layer. $T_i$ represents the conversion operation that converts the source layer to the same resolution. $\psi_t$ is used to reuse the source layer after the resolution conversion and generates a new $FRM_t$. $y_i$ represents the next pyramid feature map. $\psi_r$ is used as the fusion of the pyramid feature map of previous layer $y_{i-1}$ and $FRM_t$.

- **Conversion strategy $T_i$:** Firstly, the $1 \times 1$ convolutional layer is used to reduce the dimension of each source layer. Next, upsampling by bilinear interpolation, the scale is transformed to a scale of the same size as the convolution to be fused, thus generating

the source layer with a transformed resolution ($C2'$ and $C3'$, respectively). It is worth noting that BatchNorm normalization [31] and ReLU [32] activation function are added to every conv1 × 1 convolutional layer to handle the issue of gradient disappearance and gradient explosion during backpropagation.

- *Feature reuse $\psi_t$:* After the process of conversion strategy $T_i$, new feature maps are generated ($C2'$ and $C3'$, respectively). For reusing, there are two separate methods to merge new feature maps with $C1$, concatenation and element sum operation. Concatenation operation is often used for image detection, which can fuse the extracted convolutional features and preserve the information while increasing the dimension. The element sum operation is often used for image classification, which can increase the image information and preserve the dimension while increasing the information. Therefore, we use the concatenation operation to reuse feature information of the backbone so that the reused features are used as the input of DFF-PANet.

- *Feature fusion $\psi_r$:* After $FRM_t$ is generated, it is sent to DFF-PANet (it will be introduced in Section 3.3) with the pyramid feature map of the previous layer $y_{i-1}$ for feature fusion, and the next pyramid feature map $y_i$ is generated.

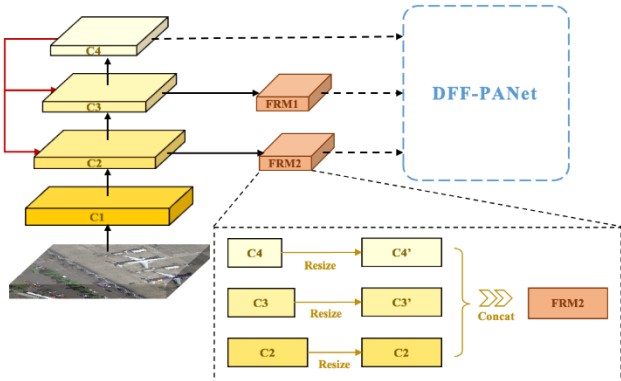

**Figure 4.** Feature Reuse Module.

### 3.3. Dense Feature Fusion Path Aggregation Network (DFF-PANet)

For pursuing higher detection accuracy, there are usually two strategies to choose from. One is to improve the backbone for feature extraction, and the other is to improve the neck for feature fusion [21]. For the strategy of enhancing the backbone, it usually leads to a large amount of computation, which limits the detection speed and makes it difficult to improve. Hence, we consider strengthening the network via improving the neck.

Most aerial objects in RSIs have different aspect ratios, but ordinary convolution cannot take full advantage of the hierarchical features in the original feature maps, thus achieving relatively low performance. For better fusing feature information from the backbone, we improved the Cross Stage Partial (CSP) module of the feature fusion network in the original YOLOv5 network structure. Inspired by reference [33], the Residual Dense Block (RDB) was used to optimize the CSP module, and the DFF-PANet was obtained. The Cross Stage Residual Dense Block (CSRDB) module is depicted in Figure 5.

As can be seen in Figure 5, the CSRDB module improved the CBS layer in the CSP module into a residual dense block, thus forming the DFF-PANet. We fully fused the feature information extracted from the backbone with the strong feature fusion ability of RDB. The network structure of RDB is depicted in Figure 6.

RDB is made up of a dense connection layer, local feature fusion and local residual learning, respectively, which constitutes a continuous memory mechanism. This mechanism is realized via transferring the state of the previous convolutional layer to the current convolutional layer [33]. Let $F_{RDB-1}$ and $F_{RDB}$ be the former and latter layers of RDB.

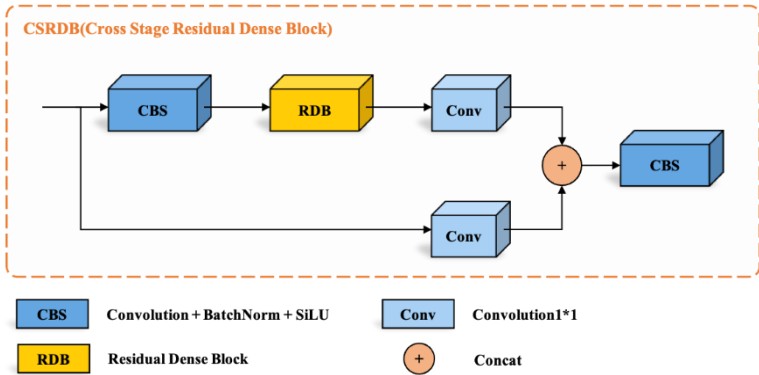

**Figure 5.** CSRDB Module.

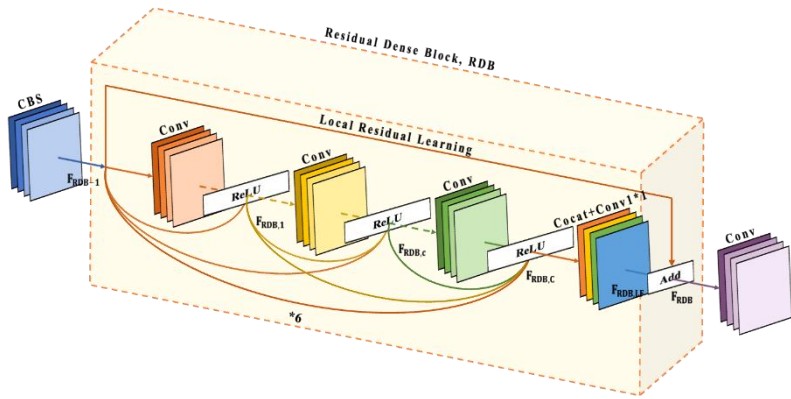

**Figure 6.** The network structure of Residual Dense Block (RDB).

- *Dense connection layer:* In this module, the dense connection layer is composed of 6 convolution layers for dense connection, with a growth rate of 32. $F_{RDB,1}$ represents the output of the first convolution. $F_{RDB,c}$ represents the output of any intermediate convolution. In this essay, $c \in \{2,3,4,5\}$. $F_{RDB,c}$ represents the output of the last convolution. In this essay, $C = 6$. Taking any intermediate convolution as an example, the output of the convolution is concatenated by the previous layer of RDB and all convolutions in RDB, then calculated by the convolutional layer and the ReLU activation function, finally, the output is obtained. It is noteworthy that all convolutions in RDB refer to the convolutions from the first convolution to the previous convolution of this convolution. Its mathematical expression can be represented as:

$$F_{RDB,c} = \sigma(W_{RDB,c}[F_{RDB-1}, F_{RDB,1}, \ldots, F_{RDB,c-1}]) \tag{3}$$

where $\sigma$ represents the ReLU activation function. $W_{RDB,c}$ represents the weight of the $c$-th convolutional layer. The dense connection layer makes CBS and the output of each layer directly connected to all subsequent layers, which not only retains feedforward features but also extracts local dense features.

- *Local feature fusion:* All features in RDB are locally fused by concatenating. In addition, the $1 \times 1$ convolutional layer is introduced to reduce the dimension and adaptively control the output information. Its mathematical expression can be expressed as:

$$F_{RDB,LF} = H_{LFF}^{RDB}([F_{RDB-1}, F_{RDB,1}, \ldots, F_{RDB,c-1}, F_{RDB,C}]) \tag{4}$$

where $H_{LFF}^{RDB}$ represents the $1 \times 1$ convolutional layer in RDB. Local feature fusion can adaptively fuse the previous convolutional features and all the convolutional features in the current RDB.

- *Local residual learning:* Local residual learning can promote the information flow between feature information before RDB and local dense features processed by RDB. The mathematical expression of the final output of RDB can be expressed as follows:

$$F_{RDB} = F_{RDB-1} + F_{RDB,LF} \tag{5}$$

where $F_{RDB,LF}$ represents the feature information after local feature fusion. Local residual learning not only contains features before RDB but also local dense features after RDB.

The RDB allows the previous convolutional layers to be directly connected to the current convolutional layer to form a continuous memory mechanism. Local feature fusion is introduced to make it learn useful local features adaptively. After gaining local dense features, global feature fusion is used to retain cumulative features and learn global features.

### 3.4. YOLO Head

YOLO Head mainly has two stages, inference and training. In inference, the model uses the trained weights to obtain the position of the bounding box. While in training, the model calculates loss and makes it decrease through repeated iterative training. When loss does not decrease any more, the training tends to be stable and better model parameters can be obtained.

#### 3.4.1. Inference

After feature fusion in the DFF-PANet, the features will be sent to YOLO Head for detection. In this thesis, we use three different detection scales to detect instances of different sizes, $32 \times 32$, $16 \times 16$ and $8 \times 8$, respectively. Taking the $8 \times 8$ detection scale as an example, the network divides the input image into $8 \times 8$ grids, each grid point is preset with three anchor boxes of different sizes. If the center of an object falls in the grid, the grid is responsible for the object. Each grid predicts three bounding boxes, each bounding box includes five parameters, $x$-coordinate, $y$-coordinate, width, height and confidence of the center point of the object, respectively. Then, the network iteratively calculates the loss value through backward propagation, constantly adjusts the properties of the anchor box and finally filters out the redundant anchor boxes by NMS. The predicted bounding box coordinates can be expressed as:

$$b_x = 2\sigma(t_x) - 0.5 + c_x \tag{6}$$

$$b_y = 2\sigma(t_y) - 0.5 + c_y \tag{7}$$

$$b_w = p_w(2\sigma(t_w))^2 \tag{8}$$

$$b_h = p_h(2\sigma(t_h))^2 \tag{9}$$

$$\sigma(x) = \frac{1}{1 + e^{-x}} \tag{10}$$

where $b_x$ and $b_y$ are the $x$ and $y$-coordinate of the center point of the prediction box. $b_w$ and $b_h$ are the width and height of the prediction box. $b_x$, $b_y$, $b_w$ and $b_h$ determine the coordinate of the prediction box. $t_x$ and $t_y$ are the offset of the object center point relative to the upper left corner of the grid where the point is located. $t_w$ and $t_h$ are the width and height of the predicted bounding box. $t_x$, $t_y$, $t_w$ and $t_h$ are the parameters obtained through iterative learning. $c_x$ and $c_y$ are the offset of the grid where the object center point is located relative to the upper left corner of the picture. $p_w$ and $p_h$ are the width and height of the anchor box. $\sigma(x)$ function is introduced to control the offset of the object center within the corresponding grid unit. The prediction box generation diagram is shown in Figure 7.

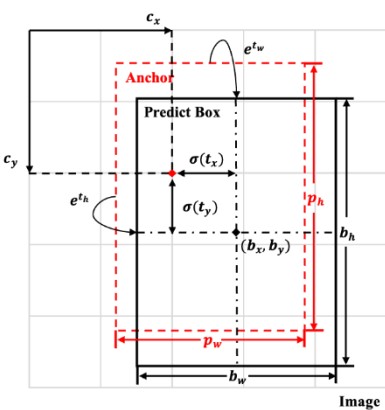

**Figure 7.** The prediction box generation diagram. The red part represents the position of the anchor box.

3.4.2. Training

For the network proposed in this thesis, the overall loss function can be expressed as:

$$Loss = \lambda_1 L_{Box} + \lambda_2 L_{Obj} + \lambda_3 L_{Cls} \tag{11}$$

where $L_{Box}$, $L_{Obj}$ and $L_{Cls}$ represent bounding box regression loss function, confidence loss function and classification loss function, respectively. Hyperparameters $\lambda_1$, $\lambda_2$ and $\lambda_3$ are default settings as $\{1, 1, 1\}$. The bounding box regression loss function is calculated by Complete Intersection over Union (CIoU), the confidence loss function and classification loss function are calculated by Binary Cross Entropy With Logits Loss (BCEWithLogitsLoss) [34]. BCEWithLogitsLoss formula is as follows:

$$BCEWithLogitsLoss = -\sum_{n=1}^{N} [x_i^* \log(\delta(x)) + (1 - x_i^*) \log(\delta(1 - x))] \tag{12}$$

where $N$ is the number of input vectors. $x_i^*$ and $x$ are the corresponding prediction vector and real vector. $\sigma(x)$ is the sigmoid function.

(1) Bounding box regression loss function
CIoU Loss [35] is introduced to calculate the position loss of the prediction box and the ground truth box. Its mathematical expression can be expressed as

$$CIoU = IoU - \frac{\rho^2(P_{box}, T_{box})}{c^2} - av \tag{13}$$

$$a = \frac{v}{1 - IoU + v} \tag{14}$$

$$v = \frac{4}{\pi^2} \left( arctan \frac{w^{gt}}{h^{gt}} - arctan \frac{w}{h} \right)^2 \tag{15}$$

$$L_{Box} = 1 - IoU + \frac{\rho^2(P_{box}, T_{box})}{c^2} + av \tag{16}$$

where $w$ and $h$ are the width and height of the prediction box $P_{box} \in R^{N_t \times (x_c, y_c, w, h)}$, respectively. $w^{gt}$ are $h^{gt}$ are the width and height of the ground truth box $T_{box} \in R^{N_t \times (x_c, y_c, w, h)}$, respectively. $N_t$ is the number of objects. $a$ is the weight coefficient. $v$ is the distance of aspect ratio between prediction box and ground truth box.

(2) Confidence loss function

$$L_{Obj} = \sum_{i}^{N_p} BCEWithLogitsLoss \left( P_{obj}, T_{obj} \right) \tag{17}$$

where $N_p$ is the number of channels of the prediction layer, the default is 3. $P_{obj} \in R^{N_p \times w_i \times h_i}$ is the prediction vector. $T_{obj} \in R^{N_p \times w_i \times h_i}$ is the real vector. $w_i(i = 1, 2, 3)$ is the width of the prediction layer. $h_i(i = 1, 2, 3)$ is the height of the prediction layer.

(3)    Classification loss function

$$L_{Cls} = \sum_i^{N_p} BCEWithLogitsLoss(P_{cls}, T_{cls}) \tag{18}$$

where $N_p$ is the channels number of the prediction layer, the default is 3. $P_{cls} \in R^{N_t \times N_c}$ is the prediction probability distribution of each category. $T_{cls} \in R^{N_t \times N_c}$ is the real probability distribution of each category. $N_t$ is the number of objects. $N_c$ is the number of categories.

### 3.5. Pseudo-Code of Network Structure

The pseudo code of the method proposed by us is shown in Algorithm 1.

---

**Algorithm 1:** A lightweight object detection method.

---

| | |
|---|---|
| **Input:** | $Input \in R^{3 \times 256 \times 256}$, *Input* refers to the input image. |
| **Step 1:** | $x = Focus(Input), x \in R^{32 \times 128 \times 128}$, *x* is sent into the backbone to gain feature maps $X = \{x_1, x_2, x_3, x_4\}$. |
| **Step 2:** | $F = \{\}$, *F* refers to the feature maps to be sent into the DFF-PANet for feature fusion. |
| | **for** *k* in range (1,4) **do** |
| | **if** $k = 1$ **then** |
| | continue |
| | **else**: |
| | if $k = 2$: $F_k \Leftarrow FRM(x_k, x_{k+1}, x_{k+2})$ |
| | if $k = 3$ : $F_k \Leftarrow FRM(x_k, x_{k+1})$ |
| | if $k = 4$ : $F_k \Leftarrow FRM(x_k)$ |
| | **end if** |
| | $F = F.append(F_k)$ |
| | **end for** |
| **Step 3:** | *F* is sent into DFF-PANet, three feature maps of different sizes $Z = \{z_1, z_2, z_3\}$ are generated. |
| **Output:** | $Results \Leftarrow Classification() \& Regression()$ |
| | **return** *Results* |

---

## 4. Experiments

We tested the proposed method on the DOTA [36] and HRSC2016 datasets [37] and compared it with other methods to evaluate the efficiency of our method. This section presents the dataset, network training, experimental results and so on.

### 4.1. Dataset

4.1.1. DOTA Dataset

The DOTA dataset [36] is a large-scale optical remote sensing dataset for object detection in aerial RSIs, which has 2806 aerial images obtained from various sensors and platforms, including 15 categories, Plane, Baseball diamond, Bridge, Ground track field, Small vehicle, Large vehicle, Ship, Tennis court, Basketball court, Storage tank, Soccer ball field, Roundabout, Harbor, Swimming pool and Helicopter, respectively. To visualize the size and location of objects in the DOTA dataset, heat maps are introduced to represent them. The heat map of the DOTA dataset distribution is shown in Figure 8. As can be seen from this figure, the dataset has objects of different sizes with uniform location distribution.

We divided the dataset into 1411 in the training set, 458 in the validation set and 937 in the test set. The labels of the dataset are composed of horizontal bounding boxes, with a total of 188,282 instances. Owing to the large images in the DOTA dataset, we cropped

the original images to $1024 \times 1024$ pixels with an overlap area of 200. After cropping the images, 15,749 images are used for training, 5297 images for validation and 10,019 images for test.

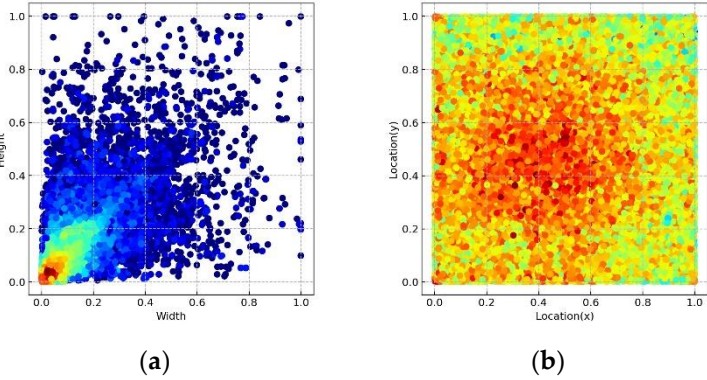

(**a**)  (**b**)

**Figure 8.** Heat maps of the DOTA dataset distribution. (**a**) The horizontal coordinate indicates the width of the object, and the vertical coordinate indicates the height of the object; (**b**) the horizontal coordinate indicates the *x*-coordinate of the object after normalization, the vertical coordinate indicates the *y*-coordinate of the object after normalization.

### 4.1.2. HRSC2016 Dataset

The HRSC2016 dataset [37] is a common dataset of optical RSIs used for ship detection. This dataset contains 1061 RSIs with a resolution ranging from 0.4 to 2 m and the image size ranging from $300 \times 300$ to $1500 \times 900$ pixels from six different ports. We divide the images into 436 training sets, 181 verification sets and 444 test sets. The labels of the dataset are composed of horizontal bounding boxes. The dataset contains a large number of ships with large aspect ratios, such as warships, aircraft carriers and cargo ships.

### *4.2. Network Training*

#### 4.2.1. Parameter Setting

The environment of this experiment is as follows. The programming language used in the experiment is Python 3.8, the model is deployed in the PyTorch 1.10.1 deep learning framework, the operating system is Ubuntu 20.04.3 LTS, the hardware platform is Inter (R) Xeon (R) Silver 4114 CPU @ 2.20 GHz with two Quadro P4000 8 GB of memory.

The pre-training parameters are illustrated as follows. The input image size is $640 \times 640$, the batch size is 16, the optimization is Stochastic Gradient Descent (SGD), the momentum is 0.9, the weight decay is 0.0005, the initial learning rate is 0.01 and the training epochs are 300 generations. We set the same parameters for other comparison methods. The initialization training parameters are shown in Table 3.

**Table 3.** The initialization training parameter.

| Input Size | Batch Size | Momentum | Weight Decay | Learning Rate | Epoch |
|---|---|---|---|---|---|
| $640 \times 640$ | 16 | 0.9 | 0.0005 | 0.01 | 300 |

#### 4.2.2. Evaluation Criteria

Precision *P*, Recall *R*, Average Precision *AP*, mean Average Precision *mAP* and *F*1-*Score* are selected to evaluate the detection ability of our proposed method quantitatively. The formula of *P*, *R* and *AP* are as follows:

$$Precision\ (P) = \frac{TP}{TP + FP} \tag{19}$$

$$Recall\ (R) = \frac{TP}{TP + FN} \tag{20}$$

$$AP = \int_0^1 P(R)dR \tag{21}$$

where $TP$ is True Positive; positive samples are predicted as positive samples. $FP$ is False Positive; negative samples are predicted as positive samples. $FN$ is False Negative; positive samples are predicted as negative samples. $AP$ is Average Precision, namely the surrounding area of the Precision-Recall curve (P-R curve), which is used to avoid the imbalance between the precision and recall, $AP$ is the value between 0 and 1. The larger the area enclosed by the P-R curve is, the better the model performance is. The $mAP$ is the average of $AP$ of all categories in the dataset. The formula is as follows:

$$mAP = \frac{1}{K} \sum_{n=1}^{K} \int_0^1 P_n(R_n)dR_n \tag{22}$$

where $K$ is the total number of the classes. $R_n$ is the recall of a given class $n$. $P_n(R_n)$ is the precision when recall of the class is $R_n$.

We also use the measurement index *F1-Score* to balance the relationship between precision and recall better. The larger the value is, the better the model performance is. The formula is as follows:

$$F1\text{-}Score = 2 \times \frac{Precision \times Recall}{Precision + Recall} \tag{23}$$

Furthermore, in order to detect the multi-scale detection capability of our method better, we also adopted COCO evaluation metrics [15], including $AP_{50}$, $AP_{75}$, $AP_S$, $AP_M$ and $AP_L$, where $AP_{50}$ is the $AP$ value when $IoU = 0.5$ ($mAP$ indicator used in this paper is the same as $AP_{50}$). $AP_{75}$ is the $AP$ value when $IoU = 0.75$. $AP_S$ is the $AP$ value of a small object ($area < 32^2$). $AP_M$ is the $AP$ value of a medium object ($32^2 < area < 96^2$). $AP_L$ is the $AP$ value of a large object ($area > 96^2$). It should be noted that in all experiments, the $IoU$ threshold was set at 0.6, which can be adjusted according to the actual application to balance false and missed detection.

### 4.3. Experimental Results

4.3.1. Experimental Results on the DOTA Dataset

We conducted some experiments on the DOTA dataset to verify the validity of our proposed method and compared our method with current popular one-stage object detection methods, such as SSD [22], YOLOv2 [11], RetinaNet [25], Adaptive Feature Aggregation Network (AFANet) [38], Self-Adaptive Anchor Selection (A2S-Det) [39] and YOLOv5, respectively, and two-stage ones, such as Rotation-Dense Feature Pyramid Network (R-DFPN) [40], Faster Region-Convolutional Neural Network (Faster R-CNN) [28], Rotation Region Proposal Networks (RRPN) [41], Image Cascade and Feature Pyramid Network (ICN) [42] and RoI Transformer (RoI Trans.) [43], respectively. To display the results better, we give each category on the dataset a corresponding name, as illustrated in Table 4.

**Table 4.** Categories on the DOTA dataset and their corresponding names.

| PL | Plane | LV | Large vehicle | SBF | Soccer ball field |
|----|-------|----|---------------|-----|-------------------|
| BD | Baseball diamond | SH | Ship | BA | Roundabout |
| BR | Bridge | TC | Tennis court | HA | Harbor |
| GTF | Ground track field | BC | Basketball court | SP | Swimming pool |
| SV | Small vehicle | ST | Storage tank | HC | Helicopter |

Table 5 displays the comparison results of object detection accuracy between our detector and other detectors on the DOTA dataset. As can be seen from the table, our method reaches the optimal result among all comparison methods, with mAP reaching 71.5%, which exceeds most single-stage and two-stage object detectors at present. As

illustrated in the table, the mAP of ours is nearly 10% higher than R-DFPN, Faster R-CNN and RRPN detection methods, small vehicle (SV) and ship (SH) are about 20% higher among categories. As the size of these categories is small, these models lack feature extraction ability for small targets, leading to poor detection performance for small targets. The model proposed by us promotes the feature extraction ability of the backbone via reusing feature information in the backbone to enhance the detection performance of the network for small and medium-sized aerial targets.

**Table 5.** The detection accuracy of each category on the DOTA dataset, the detection performance of the best is marked in red, the detection performance of the second-best is marked in green, the detection performance of the third is marked in blue.

| Model | PL | BD | BR | GTF | SV | LV | SH | TC | BC | ST | SBF | RA | HA | SP | HC | mAP |
|---|---|---|---|---|---|---|---|---|---|---|---|---|---|---|---|---|
| **Two-stage:** | | | | | | | | | | | | | | | | |
| R-DFPN [40] | 80.9 | 65.8 | 33.8 | 58.9 | 55.8 | 50.9 | 54.8 | 90.3 | 66.3 | 68.7 | 48.7 | 51.8 | 55.1 | 51.3 | 35.9 | 57.9 |
| Faster R-CNN [28] | 80.2 | 77.6 | 32.9 | 68.1 | 53.7 | 52.5 | 50.0 | 90.4 | 75.1 | 59.6 | 57.0 | 49.8 | 61.7 | 56.5 | 41.9 | 60.5 |
| RRPN [41] | 88.5 | 71.2 | 31.7 | 59.3 | 51.9 | 56.2 | 57.3 | 90.8 | 72.8 | 67.4 | 56.7 | 52.8 | 53.1 | 51.9 | 53.6 | 61.0 |
| ICN [42] | 81.4 | 74.3 | 47.7 | 70.3 | 64.9 | 67.8 | 70.0 | 90.8 | 79.1 | 78.2 | 53.6 | 62.9 | 67.0 | 64.2 | 50.2 | 68.2 |
| RoI Trans. [43] | 88.6 | 78.5 | 43.4 | 75.9 | 68.8 | 73.7 | 83.6 | 90.7 | 77.3 | 81.5 | 58.4 | 53.5 | 62.8 | 58.9 | 47.7 | 69.6 |
| **One-stage:** | | | | | | | | | | | | | | | | |
| SSD [22] | 57.9 | 32.8 | 16.1 | 18.7 | 0.1 | 36.9 | 24.7 | 81.2 | 25.1 | 47.5 | 11.2 | 31.5 | 14.1 | 9.1 | 0.0 | 29.9 |
| YOLOV2 [11] | 76.9 | 33.9 | 22.7 | 34.9 | 38.7 | 32.0 | 52.4 | 61.7 | 48.5 | 33.9 | 29.3 | 36.8 | 36.4 | 38.3 | 11.6 | 39.2 |
| RetinaNet [25] | 88.3 | 77.8 | 47.5 | 59.1 | 73.8 | 63.5 | 77.7 | 90.4 | 78.6 | 65.9 | 48.7 | 61.8 | 68.9 | 71.6 | 38.2 | 67.5 |
| AFANet [38] | 89.4 | 73.9 | 47.3 | 59.9 | 64.5 | 67.3 | 82.9 | 90.7 | 66.3 | 72.3 | 67.6 | 62.2 | 76.8 | 60.5 | 52.8 | 69.0 |
| A$^2$S-Det [39] | 89.6 | 77.9 | 46.4 | 56.5 | 75.9 | 74.8 | 86.1 | 90.6 | 81.1 | 83.7 | 50.2 | 60.9 | 65.3 | 69.8 | 50.9 | 70.6 |
| YOLOv5 | 91.6 | 75.5 | 46.0 | 61.4 | 68.1 | 85.5 | 87.8 | 93.0 | 65.9 | 69.6 | 57.2 | 58.6 | 83.9 | 61.6 | 50.6 | 70.4 |
| **Ours** | 92.1 | 73.5 | 49.0 | 63.7 | 69.1 | 85.8 | 87.9 | 93.6 | 65.9 | 71.2 | 52.6 | 61.4 | 83.5 | 63.6 | 59.0 | 71.5 |

There are fifteen classes in the DOTA dataset, the network we proposed has a detection efficiency in the top three for the most of classes. Among them, the detection efficiency of objects arranged densely is significant, for instance, Large Vehicle (LV) and Ship (SH). Secondly, the detection performance of objects in complex backgrounds has also reached good performance, such as Plane (PL) and Tennis Court (TC). Meanwhile, Helicopter (HC), which is often missed or mistakenly detected as Plane (PL), also has an accuracy about 5% higher than the sub-optimal accuracy. It is noteworthy that the detection results of these categories are all in the top three. It is because the FRM we use enables feature information in the backbone to be fully utilized, and the strong feature fusion capability of the proposed DFF-PANet handles the issue of object detection difficulties in RSIs due to excessive external interference factors to a certain extent. These experimental results indicate the availability and robustness of our method.

To better evaluate the detection validity of the method we proposed, we also drew the P-R and AP-Epoch curves to testify the availability of our proposed method. The P-R curve on the DOTA dataset is illustrated in Figure 9. We used IoU = 0.6 to calculate the precision and recall. As can be seen from the P-R curve, the detection performance of our improved method is higher than that of YOLOv5 after integrating the precision and recall. The AP-Epoch curve on the DOTA dataset is illustrated in Figure 10. We show the changes of AP within 300 epochs. As can be seen from the AP-Epoch curve, the AP of the method we put forward is higher than that of YOLOv5.

The Precision, Recall and AP values on the DOTA dataset are illustrated in Table 6. As is apparently shown in the table, compared with YOLOv5, the precision of the method we put forward is 1.8% lower, and the recall is 1.1% higher than the original. Although the precision of ours is 1.8% lower than that of the original method without improvement, the efficiency of a model cannot be evaluated only by the precision or recall, while the F1-Score can consider the relationship between precision and recall comprehensively. In terms of F1-Score, we reached 72.8%. Moreover, our improved model not only gets a better value of 71.5% on $AP_{50}$, but also increases by 1.0% on $AP_{75}$ compared with that before improvement. In $AP_S$, $AP_M$ and $AP_L$, our model increases by 0.7%, 1.3% and 1.2%, respectively, which indicates that the proposed model can strengthen the detection performance of objects of

different sizes. In terms of inference time, our method has increased from 3.3 to 4.6 ms. Although the inference speed has slowed down, it still meets the requirement of real-time detection (more than 30 frames; that is, the inference time is less than 33.3 ms) [44]. It shows that the proposed method not only does not bring much burden to the detection speed but also improves the detection performance and achieves a better balance between speed and accuracy.

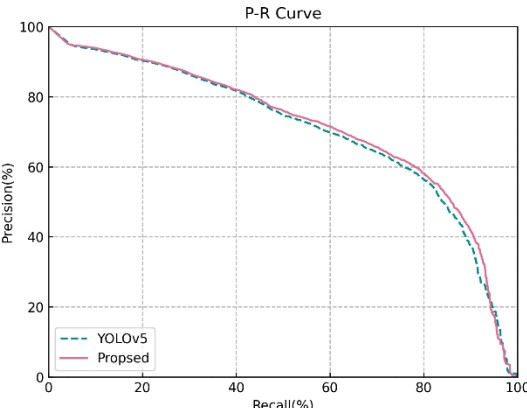

**Figure 9.** P-R Curve on the DOTA dataset.

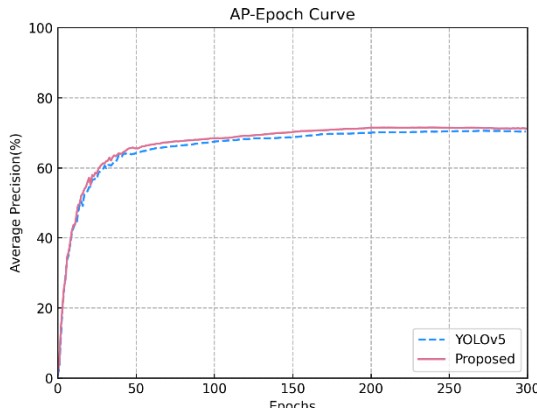

**Figure 10.** AP-Epoch Curve on the DOTA dataset.

**Table 6.** Precision, Recall and AP values.

| Method | Precision and Recall (%) | | | AP Values (%) | | | | | Times (ms) |
|---|---|---|---|---|---|---|---|---|---|
| | Precision | Recall | F1-Score | $AP_{50}$ | $AP_{75}$ | $AP_S$ | $AP_M$ | $AP_L$ | - |
| YOLOv5 | 79.0 | 65.9 | 71.9 | 70.4 | 44.9 | 17.7 | 41.7 | 55.6 | 3.3 |
| Proposed | 77.2 | 67.0 | 72.8 | 71.5 | 45.9 | 18.4 | 43.0 | 56.8 | 4.6 |

### 4.3.2. Experimental Results on the HRSC2016 Dataset

Table 7 shows the comparison results of object detection accuracy between our proposed method and other methods on the HRSC2016 dataset. In the methods we compare, single-stage object detection methods include Rotation Region Proposal Networks (RRPN) [41], Rotated Region Proposal and discrimination Networks ($R^2$PN) [45], RoI Transformer (RoI Trans.) [43], Densely Coded Labels (DCL) [46] and YOLOv5. Two-stage object detection methods include RetinaNet [25], Rotation-sensitive Regression Detector (RRD) [47], Rotation Sensitive Detector (RSDet) [48], Dynamic Anchor Learning (DAL) [49], Refi Ned Single-Stage Detector (R3Det) [50] and RepVGG-YOLO [51]. It can be seen from the table that the method we proposed achieves the best result among all the comparison methods, with mAP reaching 93.3%, which is 1.8 percent higher than the suboptimal method (RepVGG-YOLO). The experimental results show that the proposed method can

achieve better detection results, even for ship targets with a large aspect ratio. Secondly, although the inference time (4.0 ms) of our proposed model is lower than that of YOLOv5, our method is far superior to that of other methods and has a great advantage in precision, so it is acceptable to rank second in speed. Table 7 shows that the proposed method meets both the accuracy and speed requirements for object detection.

**Table 7.** The detection accuracy on the HRSC2016 dataset.

| Model | mAP | Inference Time (ms) |
|---|---|---|
| Two-stage: | | |
| RRPN [41] | 79.1 | 285.7 |
| R$^2$PN [45] | 79.6 | - |
| RoI Trans. [43] | 86.2 | 166.7 |
| DCL [46] | 89.5 | - |
| One-stage: | | |
| RetinaNet [25] | 80.8 | - |
| RRD [47] | 84.3 | - |
| RSDet [48] | 86.5 | - |
| DAL [49] | 89.0 | 29.4 |
| R3Det [50] | 89.3 | 83.3 |
| RepVGG-YOLO [51] | 91.5 | 45.5 |
| YOLOv5 | 92.4 | 2.9 |
| Ours | 93.3 | 4.0 |

At the same time, the P-R and Loss curves are also drawn to verify the effectiveness of our proposed method. The P-R curve on the HRSC2016 dataset is shown in Figure 11. It can be seen from the P-R curve that the curve of the proposed method is always higher than YOLOv5, indicating that the detection accuracy of the model is superior to YOLOv5. The Loss curve on the HRSC2016 dataset is shown in Figure 12. We show the changes of the loss value within 300 epochs. It can be seen from the Loss curve that the curve shows a downward trend and eventually converges to a certain range. Secondly, the convergence rate of the improved model is faster than YOLOv5, indicating that our method has better detection performance.

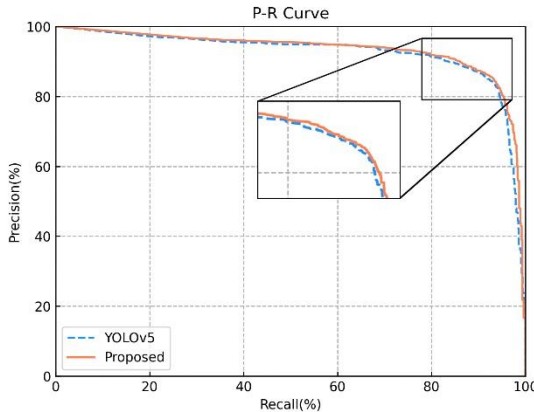

**Figure 11.** P-R Curve on the HRSC2016 dataset.

### 4.4. Visualization Results

The visualization results of the model on the DOTA dataset are shown in Figure 13. Figure 13(a1–a4) displays the objects with a small size; Figure 13(b1–b4) reviews the objects with different sizes in RSIs; Figure 13(c1–c4) demonstrates the instances in complex backgrounds; Figure 13(d1–d4) shows the detection effect of densely arranged objects. As is apparently shown in Figure 13, the network proposed by us has achieved good detection performance on small instances, objects of different sizes, instances in complex backgrounds and objects in dense scenes.

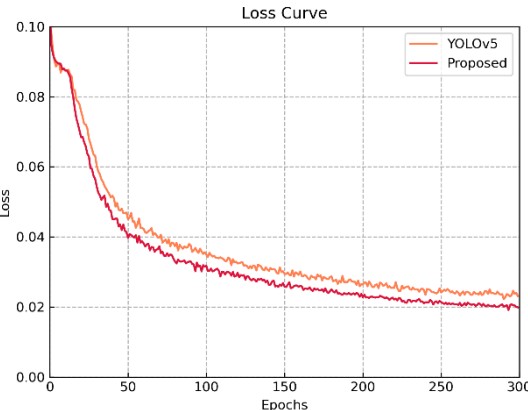

**Figure 12.** Loss Curve on the HRSC2016 dataset.

| | | | |
|---|---|---|---|
| (**a1**) | (**a2**) | (**a3**) | (**a4**) |
| (**b1**) | (**b2**) | (**b3**) | (**b4**) |
| (**c1**) | (**c2**) | (**c3**) | (**c4**) |
| (**d1**) | (**d2**) | (**d3**) | (**d4**) |

**Figure 13.** Visualization results in four cases of the DOTA dataset. (**a1–a4**) Shows the objects with a small size; (**b1–b4**) shows the objects with different sizes in RSIs; (**c1–c4**) shows the objects in complex backgrounds; (**d1–d4**) shows the objects arranged densely.

The visualization results of our proposed model on the HRSC2016 dataset are shown in Figure 14. As can be seen from the figure, even for ship objects placed side by side and under complex backgrounds, our model can perform accurate detection and finally output prediction results of high quality.

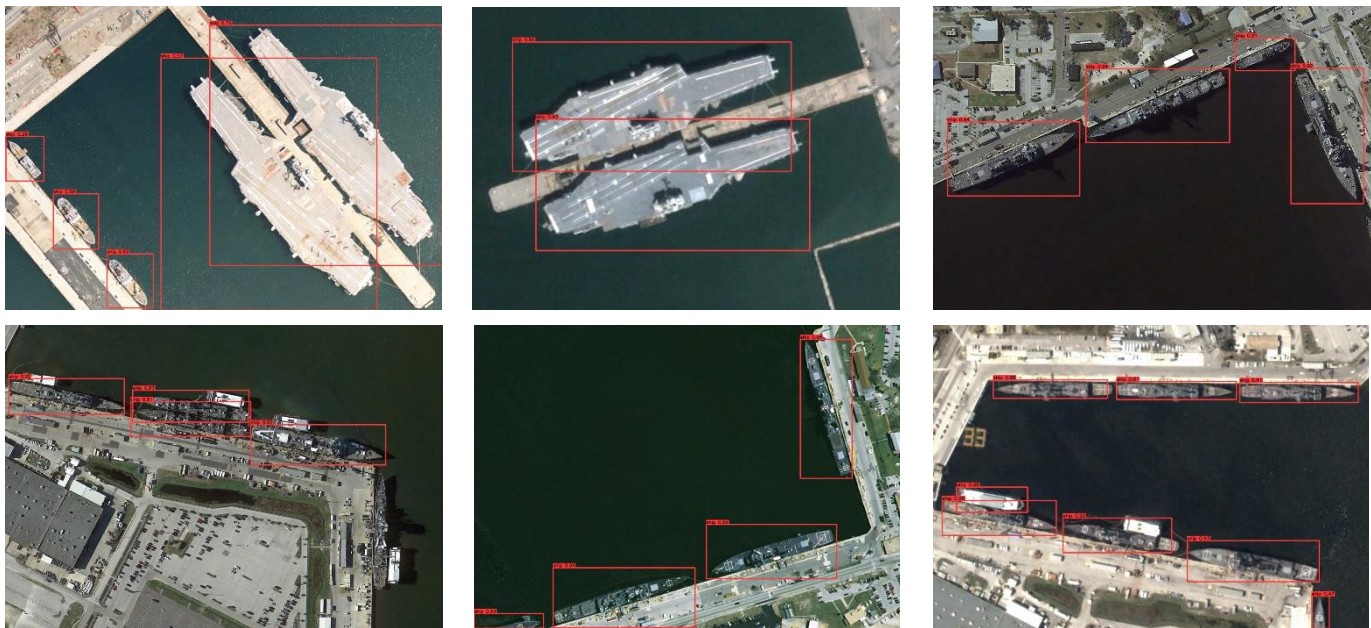

**Figure 14.** Visualization results on the HRSC2016 dataset.

We also compared our proposed method with the original YOLOv5. Figure 15 shows the visualization results of the comparison of the datasets. Figure 15(a1–a4) Represents the original image; Figure 15(b1–b4) represents the experimental results of YOLOv5; Figure 15(c1–c4) represents the experimental results of the method raised by us. As can be seen from Figure 15, to some extent, the model raised by us can not only solve the problem of small instances but also solve the problem of false and missed detection caused by shadow, similar objects, complex backgrounds and side-by-side placement. While facing the above-mentioned situations, the original YOLOv5 sometimes produces false or missed detection.

*4.5. Ablation Study*

To better testify the performance of our proposed modules, we tested each module through the ablation studies. They used the same hyperparameters and parameter settings. All experiments were tested on the same dataset, and the ablation studies on the DOTA dataset are shown in Table 8. We used Params, Floating Point Operations (FLOPs), Precision, Recall, mAP and F1-Score to verify the availability of the module we proposed.

In Table 8, we used YOLOv5 as the baseline and obtained 70.4% mAP without adding the FRM and the DFF-PANet.

- **Feature Reuse Module (FRM):** To demonstrate the validity of the FRM, we added the FRM based on the baseline. With the help of the FRM, the network arrived at 70.8% mAP, which was 0.4% higher than the baseline. Moreover, the experimental result was higher than those without the FRM. It is because before using FRM, low-level feature maps lack rich semantic information, which leads to insufficient detection ability of small instances While adding FRM, the position information in low-level feature maps can fully mix with semantic information in high-level ones, thereby enhancing the feature reuse ability of the backbone to promote the problem of insufficient feature extraction ability of the network.

- **Dense Feature Fusion Path Aggregation Network (DFF-PANet):** To certify the validity of the DFF-PANet, the neck of the baseline was replaced by the DFF-PANet. As is apparently shown in the table, the network reached 71.3% mAP, which was 0.9% higher than the baseline after adding the DFF-PANet. It is because of the strong feature fusion ability of residual dense blocks in the DFF-PANet. After obtaining the local dense features, it retains the accumulated feature information through global feature fusion to improve the network performance.
- **Proposed Method:** When both the FRM and the DFF-PANet were added to the model, the method we put forward was obtained. We reached 71.5% mAP, which was 1.1% higher than the baseline. Our improved method also reached the highest F1-Score. It displays that the FRM and the DFF-PANet are both effective modules to improve the network performance; they both enhance the detection ability of the model to a certain extent.

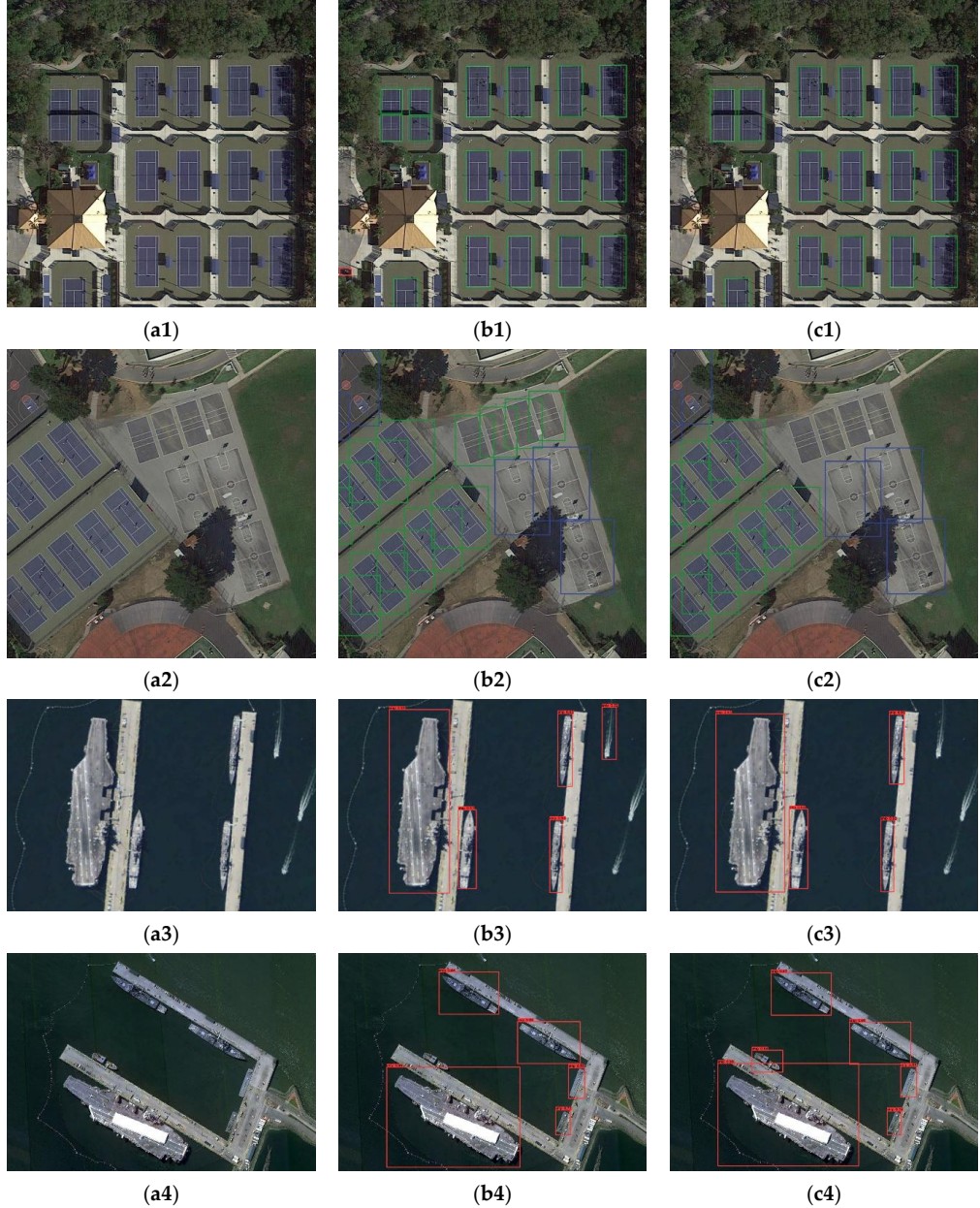

**Figure 15.** Comparison diagram on the DOTA dataset and the HRSC2016 dataset. (**a1**–**a4**) Represents the original image; (**b1**–**b4**) represents the experimental results of YOLOv5; (**c1**–**c4**) represents the experimental results of the network structure proposed by us.

**Table 8.** Ablation results.

| Model | FRM | DFF-PANet | Params (M) | FLOPs (B) | Precision (%) | Recall (%) | F1-Score (%) | mAP@.5 (%) |
|---|---|---|---|---|---|---|---|---|
| Baseline | - | - | 7.1 | 16.4 | 79.0 | 65.9 | 71.9 | 70.4 |
| A | √ | | 8.4 | 17.8 | 75.0 | 68.2 | 71.4 | 71.0 (+0.6) |
| B | | √ | 7.8 | 20.9 | 80.4 | 65.8 | 72.4 | 71.3 (+0.9) |
| C | √ | √ | 9.2 | 22.2 | 77.2 | 67.0 | 72.8 | 71.5 (+1.1) |

## 5. Discussion

In this section, we will discuss the lightweight remote sensing object detection method proposed in this thesis from three aspects: (1) Model method; (2) Lightweight model; (3) Model accuracy.

- **Model method:** We compared our proposed model with different versions of YOLOv5 models on the DOTA datasets, namely, YOLOv5n (nano), YOLOv5s (small) and YOLOv5m (medium). The experimental results are shown in Table 9. As can be seen from Table 9, the method proposed by us has certain improvements on different versions of YOLOv5 models, which increased 1.6%, 1.1% and 0.9%, respectively.

- **Lightweight model:** Currently, designing a network structure that can balance detection accuracy and model parameters at the same time is the mainstream direction in object detection algorithms. Although most network structures achieve high accuracy, they usually require a large amount of calculation, and it is difficult to achieve good detection performance with a small amount of calculation. In this study, the YOLOv5s model used by us achieves a balance between detection accuracy and model parameters. The model parameters are only 9.2 M, and the inference time is 4.6 ms, meeting the requirements of real-time detection (more than 30 frames; that is, the inference time is less than 33.3 ms). Therefore, it can be deployed on front-end devices, such as mobile terminals [52]. Table 9 shows that the number of parameters in YOLOv5s is nearly 13 M less than that in YOLOv5m, which greatly reduces the model parameters. Compared with YOLOv5n, although the model parameters are 6.1 M more than it, the detection accuracy is improved by 3%. Therefore, compared with YOLOv5n, the increased number of parameters is acceptable.

- **Model accuracy:** Comparative analysis of the dataset and the ablation experiments mentioned above shows that our proposed method has excellent performance for instances of different sizes or with many external interference factors. However, as can be seen from the data in Table 5, the detection accuracy of objects, such as Ground track field (GTF), Basketball court (BC) and Soccer ball field (SBF), still lags behind first place. Our method does not achieve a satisfactory result when dealing with such objects. It may be because such objects are sometimes in the same background, and their texture information is similar; the feature information cannot be clearly identified by the model, leading to the low detection performance of objects. In future work, we hope to improve the model in this aspect.

**Table 9.** Experimental results of different models on the DOTA dataset (* represents the model proposed in this paper).

| Model | Size (Pixels) | Time (ms) | Params (M) | FLOPs (B) | mAP (%) |
|---|---|---|---|---|---|
| YOLOv5n | 640 | 2.1 | 1.9 | 4.7 | 66.9 |
| YOLOv5n + FRM + DFF-PANet | 640 | 2.9 | 3.1 | 9.0 | 68.5 |
| YOLOv5s | 640 | 3.3 | 7.1 | 16.4 | 70.4 |
| **YOLOv5s + FRM + DFF-PANet (*)** | 640 | 4.6 | 9.2 | 22.2 | 71.5 |
| YOLOv5m | 640 | 7.6 | 21.2 | 51.4 | 72.4 |
| YOLOv5m + FRM + DFF-PANet | 640 | 8.9 | 22.2 | 53.5 | 73.3 |

## 6. Conclusions

In this article, we found the following difficulties in RSIs. First, the size of remote sensing targets is usually very small compared with the imagery. Second, RSIs are often disturbed by external factors, such as shadows, similar objects and complex backgrounds. Third, objects side by side in RSIs lead to a high rate of missed detection. To deal with the difficulties and consider the accuracy and speed of detection, this paper proposes a lightweight object detection method based on RSIs.

(1) First, we use the Feature Reuse Module (FRM) to reuse feature maps in the backbone; this module can enhance the detection ability of the network for small and medium-sized targets via fusing semantic information and location information.

(2) After that, we designed the Dense Feature Fusion Path Aggregation Network (DFF-PANet) to better handle the issue of external interference factors in RSIs.

Experiments on the dataset demonstrate that compared with other algorithms, our method obtains 71.5% mAP, an improvement of 1.1%, as well as exceeding most of the current single-stage and two-stage detection methods. The method we raised has good performance in multi-scale remote sensing object detection. As can be seen from the visualization results, the model raised in this paper can achieve good performance.

However, some anchor boxes may be filtered when facing rotating objects on account of the use of horizontal anchor boxes, thus increasing false and missed detection of some objects. In the future, we will introduce rotating anchor boxes to further strengthen the detection performance of the model.

**Author Contributions:** Conceptualization, Liming Zhou; methodology, Xiaohan Rao; software, Xiaohan Rao; validation, Liming Zhou, Xiaohan Rao, Yahui Li and Yinghao Lin; formal analysis, Yinghao Lin; resources, Xianyu Zuo; writing—original draft preparation, Liming Zhou and Xiaohan Rao; writing—review and editing, Liming Zhou and Baojun Qiao; visualization, Baojun Qiao; supervision, Yahui Li; funding acquisition, Yinghao Lin. All authors have read and agreed to the published version of the manuscript.

**Funding:** This work was supported by grants from the National Basic Research Program of China (Grant number 2019YFE0126600); the Major Project of Science and Technology of Henan Province (Grant number 201400210300); the Key Scientific and Technological Project of Henan Province (Grant number 212102210496); the Key Research and Promotion Projects of Henan Province (Grant numbers 212102210393; 202102110121; 222102320163); and Kaifeng science and technology development plan (Grant number 2002001).

**Institutional Review Board Statement:** Not applicable.

**Informed Consent Statement:** Not applicable.

**Data Availability Statement:** The data used to support the findings of this study are available from the corresponding author upon request.

**Acknowledgments:** We sincerely thank the anonymous reviewers for the critical comments and suggestions for improving the manuscript.

**Conflicts of Interest:** The authors declare no conflict of interest.

## Abbreviations

The abbreviations used in this thesis are as follows:

| | |
|---|---|
| $A^2$S-Det | Self-Adaptive Anchor Selection |
| AFANet | Adaptive Feature Aggregation Network |
| BCEWithLogitsLoss | Binary Cross Entropy With Logits Loss |
| CF2PN | Cross-Scale Feature Fusion Pyramid Network |
| CIoU | Complete Intersection over Union |
| CNNs | Convolutional Neural Networks |
| CSPDarknet53 | Cross Stage Partial Darknet 53 |
| CSRDB | Cross Stage Residual Dense Block |

| DFF-PANet | Dense Feature Fusion Path Aggregation Network |
|-----------|------------------------------------------------|
| DOTA | Dataset of Object deTection in Aerial images |
| DPM | Deformable Parts Model |
| FLOPs | Floating Point Operations |
| FRM | Feature Reuse Module |
| HOG | Histogram of Oriented Gradients |
| ICN | Image Cascade and Feature Pyramid Network |
| IoU | Intersection over Union |
| M2Det | Multi-level and Multi-scale Detector |
| MFPNet | Multi-Feature Pyramid Network |
| MS COCO | Microsoft Common Objects in Context |
| MSE-DenseNet | Multi-scale SELU DenseNet |
| NMS | Non-Maximum Suppression |
| Pascal VOC | Pascal Visual Object Classes |
| P-R curve | Precision-Recall curve |
| R-CNN | Region-Convolutional Neural Network |
| R-DFPN | Rotation-Dense Feature Pyramid Network |
| RDB | Residual Dense Block |
| RoI | Region of Interest |
| RoI Trans. | RoI Transformer |
| RRPN | Rotation Region Proposal Networks |
| RPN | Region Proposal Networks |
| RSIs | Remote Sensing Images |
| SGD | Stochastic Gradient Descent |
| SSD | Single Shot MultiBox Detector |
| SVM | Support Vector Machine |
| YOLO | You Only Look Once |

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
