# Peer review of "A Lightweight Object Detection Method in Aerial Images Based on Dense Feature Fusion Path Aggregation Network"

_ijgi, doi:10.3390/ijgi11030189_

Round 1

Reviewer 1 Report

The presented work focuses on a new object detection from aerial images based on dense feature path aggregation network. This is a interesting work and the research and the manuscript are structured well.

However, here reflections:

  • If it is possible, reduce the introduction chapter;
  • May be interesting a discussion chapter (before conclusions) with critical considerations.

thank you

Reviewer 2 Report

Excellent paper. Only minor formatting issues are asked to change as indicated in the attached docx document 

Reviewer 3 Report

The authors proposed a DFF-PANet by integrates dense featue fusion for objects detection the experimental results demonstrated that the proposed method is effective and efficiency.

Here are some minor concerns:

  1.  The authors should add some comments about the transferbility of the proposed method

Reviewer 4 Report

The paper proposes an object detection method in Remote Sensing Images (RSI). The authors fused semantic and location information in feature maps by Feature Reuse Module which can enrich feature information extracted from the backbone. They designed a Dense Feature Fusion Path Aggregation Network by using Cross Stage Residual Dense Block. It can handle the problem of external interference caused by complex and changeable RSIs better. The topic is interesting and matches well for MDPI Machines journal. The paper contains meaningful review of related works. However the paper has some unclear points and the following major concerns.

1. On the one hand, the authors claim that

«We performed experiments on the Dataset for Object Detection in Aerial Images (DOTA) dataset, the accuracy of ours reached 71.5% mAP, which exceeds most object detectors of one-stage and two-stage at present.»

The set of standard Yolo models, in addition to the small model (7.2M params), also contains a nano model (1.9M params) and a medium model (21.2M params). Is the improvement in accuracy achieved over the small Yolo model systematic? Or will applying the proposed method to other standard Yolo models give relatively better or worse results?

2. On the other hand, the authors note that

«Meanwhile, the size of our model is only 9.2M, which satisfies the requirement of lightweight.»

What is meant by lightweight model? For which devices is this model lightweight? What are the parameters of these devices? Moreover, the inference time on these devices for standard Yolo models and for proposed models should be given.

The goal of the authors is to «consider the accuracy and speed of detection». Thus, they must show that the solution they have chosen is optimal among other solutions.

3. There are also typos in the paper.

Some period signs are missing from Table 1.

Round 2

Reviewer 4 Report

The authors of the paper have made corrections in accordance with my comments.

This manuscript is a resubmission of an earlier submission. The following is a list of the peer review reports and author responses from that submission.